# Genomic analysis of *Acinetobacter baumannii* DUEMBL6 reveals diesel bioremediation potential and biosafety concerns

Nizam Uddin[1,2]☯, Monjima Islam Prova[1]☯, Muttasim Billaha[1]☯, Tasnimul Arabi Anik 🔘[1]☯, Rahat Uzzaman[1]☯, Nadia Haider[1]☯, Faruk Islam[1]☯, Humaira Akhter[1]☯, Anowara Begum 🔘[1]*

**1** Environmental Microbiology Laboratory, Department of Microbiology, University of Dhaka, Dhaka, Bangladesh, **2** Bangladesh Council of Scientific and Industrial Research, Dhaka, Bangladesh

☯ These authors contributed equally to this work.
* anowara@du.ac.bd

## Abstract

This study investigates the bioremediation potential of bacterial isolates from diesel-contaminated soils in Dhaka, Bangladesh. A total of 34 morphologically distinct bacterial strains were isolated from hydrocarbon-polluted sites, with *Acinetobacter baumannii* emerging as the dominant species (41.2%), followed by *Pseudomonas otitidis* and *Klebsiella pneumoniae* (11.8% each). Genetic screening revealed that 32.35% of isolates harbored the *alkB* gene (alkane hydroxylase), while 58.82% carried *catE* (catechol-2,3-dioxygenase), indicating a strong predisposition for aromatic hydrocarbon degradation. Following turbidimetric screening of 34 bacterial isolates, 11 demonstrating superior growth were selected for gravimetric degradation assessment. Among these, isolate MB 1002 (*Pseudomonas nitroreducens*) demonstrated the highest degradative capability at 46.92%, followed by MB 751 (*Acinetobacter baumannii*) at 41.18% and MB 750 (*Pseudomonas aeruginosa*) at 39.16%. The MB 1002 and MB 751 both were positive for *alkB.* FTIR analysis revealed that both MB 751 and MB 1002 contribute to diesel degradation, with MB 751 showing stronger oxidation patterns due to the presence of carboxyl functional groups. Due to its superior oxidative capability, MB 751 (designated *A. baumannii* DUEMBL6) was selected for whole-genome sequencing. The sequence analysis of *A. baumannii* DUEMBL6 (deposited as JBLODW000000000) revealed: (1) hydrocarbon degradation genes (*alkB*, *ssuD*, catechol dioxygenases); (2) 7 biosynthetic gene clusters including siderophores (100% similarity to baumannoferrin); and (3) complete xenobiotic degradation pathways for aliphatic/aromatic compounds. Despite its bioremediation potential, *A. baumannii* DUEMBL6 harbored 26 antibiotic resistance genes (e.g., *blaOXA-338*, *adeABC* efflux pumps) and 33 virulence factors (e.g., *csu* pilus, biofilm genes), with an 86.1% pathogenicity probability. These findings highlight *A. baumannii*

**Data availability statement:** All relevant data are within the paper and its Supporting information files. Additionally, the whole genome sequence of A. baumannii DUEMBL6 has been deposited in the DDBJ/ENA/GenBank databases under BioProject PRJNA1217114, BioSample SAMN46479258, and Accession no. JBLODW000000000.

**Funding:** The research was funded by the Ministry of Science and Technology, and University Grant Commission, Bangladesh. The funders had no role in study design, data collection and analysis, decision to publish, or preparation of the manuscript.

**Competing interests:** The authors have declared that no competing interests exist.

DUEMBL6's dual role as a promising bioremediation agent and a potential public health risk, necessitating careful strain selection for environmental applications.

## Introduction

Petroleum hydrocarbons remain indispensable to global energy systems despite international efforts to transition toward renewable alternatives. The United Nations and International Energy Agency (IEA) have advocated for reduced crude oil dependence through biofuel adoption (IEA, 2022), yet petroleum products continue to dominate energy markets, particularly in developing economies. From running industries to managing household chores, crude oil in various forms like petrol, gasoline, diesel, and kerosene is widely used. Crude oil is a well-known environmental pollutant, threatening the lives of marine and soil ecosystems. Once introduced into the environment, as the oil starts weathering, it produces recalcitrant and xenobiotic molecules such as polycyclic aromatic hydrocarbons (PAHs), halogenated and nitro-aromatic compounds, pentachlorophenol, benzene, toluene, etc., which alter the physicochemical properties of local habitats, plants, and the microbial consortium of that environment [1–4]. Certain hydrocarbons found in oil, specifically aromatic compounds, possess toxicity [5] and exhibit resistance to degradation, resulting in their persistence in soil [6]. As a consequence, the ecology is greatly impacted. Oxygen and water deficiency, along with nitrogen and phosphorus scarcity, restrict the growth of living entities. The microbial ecology changes as many microbes gradually vanish due to nutrient deficiency, while others capable of degrading oil thrive in that environment, ultimately altering the composition of soil or water. This phenomenon is particularly concerning in the case of oil spills, which often occur accidentally during transportation. Many countries, including Bangladesh, have witnessed major oil spills in ecological hotspots. In 2014, due to the collision between an oil-containing tank and a vessel, 350,000 liters of black furnace oil spilled into a river [7] in a restricted area of the Sundarbans, reportedly disturbing the local ecology of that environment.

These oil spill events are becoming more frequent, with rising energy demands. Particularly, the demand for diesel and heating oil in Bangladesh has reached record highs in recent years. In 2023, the country consumed approximately 95.49 thousand barrels per day of diesel and heating oil, up from 90.19 thousand barrels per day in 2022. This marks the highest level ever recorded for Bangladesh, with the long-term average from 1980 to 2023 being 41.5 thousand barrels per day [8]. Diesel is the dominant liquid fuel in the country, accounting for the largest share of petroleum product use, particularly in the transport sector (62%), followed by agriculture (17%), power generation (13%), and industry (5%) as of the 2021−22 fiscal year [9]. As a refined crude oil product, diesel consists mainly of naphthalene and paraffinic hydrocarbons and is a major pollutant of soil and water [10,11]. Its environmental consequences include reduced plant growth, impaired germination, and poor soil aeration due to clogging of pore spaces [12].

While physical and chemical techniques are employed for cleaning up spilled oil, these methods are often costly and impractical in low-and-middle-income countries (LMIC) like Bangladesh. Therefore, bioremediation is recommended over the former methods. Bioremediation facilitates the degradation of oil spills through the interaction of microorganisms, which utilize their metabolic and enzymatic properties. Microorganisms typically contain oil-degrading genes that produce enzymes for various metabolic pathways, converting the oil into a soluble form. Bioaugmentation and biostimulation are key factors here, as microbes act as natural supplements to the environment. Various bacteria such as *Pseudomonas*, *Acinetobacter*, *Rhodococcus*, *Klebsiella*, and *Nocardia* have different approaches to degrading oil components. *Rhodococcus* has previously been documented as an efficient diesel oil degrader [13,14]. Some of these bacteria employ adaptive strategies such as biofilm formation and biosurfactant production, which function synergistically to reduce interfacial tension and increase hydrocarbon bioavailability [15], while simultaneously protecting microbial communities from environmental stressors [16].

Given Bangladesh's escalating diesel consumption and vulnerability to oil spills, this study sought to evaluate the bioremediation potential of indigenous soil microorganisms from contaminated sites in Dhaka. We employed a multi-method approach to: (1) isolate and characterize native diesel-degrading bacteria, (2) quantify their degradation efficiency through gravimetric and FTIR analyses, and (3) genomically profile the most promising isolate for both metabolic potential and biosafety risks.

## Materials and methods

### Sample collection and processing

Samples of soil contaminated with petroleum hydrocarbons were collected from four filling stations in Dhaka (23° 43' 56.43'' N/90° 23' 7.51'' E, 23° 45' 38.43'' N/90° 22' 23.69'' E, 23° 47' 18.68'' N/90° 19' 33.38'' E, and 23° 50' 2.03'' N/90° 15' 35.01'' E). The filling stations were selected based on higher usage and greater hydrocarbon contamination observed at these sites. A total of 40 soil samples were collected from four filling stations in Dhaka, with 10 samples from each location. The soil samples were taken from a depth of 5–10 cm in areas with potential fuel spillage and near fuel dispensers. The samples were collected in sterile bottles to avoid contamination and were immediately transported to the laboratory in controlled conditions. Upon arrival, the samples were processed promptly to prevent any alteration in the soil's microbial community. One gram of soil from each sample was inoculated into 100 mL of freshly prepared Bushnell Haas medium (BHM) supplemented with 1% diesel in a sterile flask. The cultures were incubated at 30°C with shaking at 180 rpm for 7 days to enrich diesel-degrading bacteria. After incubation, 1 mL of each culture was centrifuged at 14000 rpm for 10 minutes to pellet the cells, and the diesel-containing supernatant was discarded. The pellet was resuspended in 1 mL of sterile saline (0.85% NaCl), and the optical density (OD) of the suspension was measured at 600 nm using sterile saline as the blank. Cultures with an $OD_{600} > 0.5$ were serially diluted ($10^{-1}$ to $10^{-6}$) in sterile saline, and 100 µL of each dilution was spread onto BHM agar plates supplemented with 1% (w/v) diesel. Plates were incubated at 30°C for 5–7 days, and morphologically distinct colonies were selected and streaked onto fresh BHM-diesel agar plates to obtain pure cultures.

### Identification of isolates

The isolated pure bacterial cultures were identified via 16S rRNA gene sequencing. Genomic DNA was extracted using the QIAamp DNA Mini Kit (QIAGEN, Germany) and quantified with a Colibri Microvolume Spectrometer (Titertek-Berthold, Germany). Universal primers 16S F (5'-AGAGTTTGATCCTGGCTCAG-3') and 16S R (5'-TACCTTGTTTTACGACTC-3') [17] were used for PCR amplification. Reactions were run in a Thermo Fisher Veriti 96-well Thermal Cycler with 35 cycles at 94 °C (1 min), 47 °C (1 min), and 74 °C (1 min), followed by a final extension at 72 °C for 10 minutes.

### Determination of biofilm-forming ability

Biofilm production was assessed through complementary qualitative and quantitative approaches. Qualitative screening utilized Congo Red Agar (CRA) plates containing Brain Heart Infusion agar (37 g/L), sucrose (50 g/L), and filter-sterilized

Congo Red (0.8 g/L), inoculated with fresh bacterial cultures and incubated at 30°C for 48 hours; biofilm-forming isolates were identified by black, dry, crystalline colonies indicative of exopolysaccharide (EPS) production, while non-producers appeared pink/red with smooth morphology. For quantitative analysis, biofilm biomass was measured via a standardized microtiter plate assay [18], where bacterial suspensions (OD (600 nm) = 0.1 in Bushnell Haas Broth + 1% diesel) were incubated statically in 96-well polystyrene plates at 30°C for 48 hours. After removing non-adherent cells through PBS washing, biofilms were fixed with methanol, stained with 0.1% crystal violet, and solubilized in 33% glacial acetic acid for spectrophotometric quantification at 600 nm. The optical density cut-off value (ODc) was set as three standard deviations (SD) above the mean of the optical density (OD) of the negative control as shown in the following formula: ODc = average OD of negative control + (3 × SD of negative control). The optical density (OD) was determined by calculating the mean of all the replicates. The results were summarized into four categories based on their optical densities: (1) strong biofilm producer (4 × ODc < OD); (2) medium biofilm producer (2 × ODc < OD ≤ 4 × ODc); (3) weak biofilm producer (ODc < OD ≤ 2 × ODc); and (4) non-biofilm producer (OD ≤ ODc).

### Detection of oil-degrading genes by PCR

Two oil-degrading genes, including, *alkB* and *catE* were detected in the isolates by PCR using primers listed in Table 1. PCR cycling conditions were initial denaturation 95 °C (5 min), followed by 35 cycles of denaturation 95 °C (15 s), $T_m$ for annealing (15 s) and extension at 72 °C (1 min). Final extension at 72 °C for 5 min. PCR products were visualized using Gel Doc (BioRad, USA) following electrophoresis through 1.5% agarose gel stained with ethidium bromide.

### Determination of bacterial biodegradation activity by turbidimetric method

For biodegradation activity assessment, pure cultures were inoculated into 5 mL of Bushnell Haas Broth (BHB) supplemented with 1% (w/v) diesel and incubated at 30°C with shaking at 180 rpm for 15 days. Bacterial growth was monitored by measuring OD at 600 nm every 2 days after centrifuging 1 mL of culture at 14,000 rpm for 10 minutes, discarding the supernatant, and resuspending the pellet in 1 mL sterile saline to remove diesel interference.

### Determination of diesel oil degradation using the gravimetric analysis

The level of diesel degradation was determined using the gravimetric analysis [21]. Initially, bacterial cultures were inoculated in 100 ml of Bushnell-Haas broth supplemented with 1% diesel oil as a carbon source in an Erlenmeyer flask and incubated at 30°C for 7 days. After incubation, the cell pellet was separated by centrifugation at 14000 rpm for 10 minutes, resulting in a pellet of cell debris and a supernatant containing residual oil. The remaining diesel was extracted using hexane. Equal volumes of hexane and the cell-free supernatant were mixed to separate the diesel from the aqueous phase, which remained in the organic phase of hexane. The organic layer was then transferred to a separate falcon tube. For oil degradation estimation, 5 ml of n-hexane was added to the original flasks, and the contents were transferred to a separating funnel. This extraction was performed three times to ensure complete recovery of the oil. The hexane containing the remaining oil was left in a fume hood. The amount of residual oil was measured after extraction of oil from the medium. Negative controls (BHB + 1% diesel without bacteria) and positive controls (*Pseudomonas putida*) were included to

Table 1. Primer sequences for amplification of oil-degrading genes.

| Target gene | Primer Name | Primer Sequences (5'-3') | Amplicon size (bp) | (Ta) | Ref. |
|---|---|---|---|---|---|
| *alkB* | alkB1a- le | AAAATGGAGGACGGTCG ATA | 330 bp | 55°C | [19] |
| | alkB1- Ri | ACTCGAACTATGACCA GCAGTACTAGTGCCAAA | | | |
| *catE* | catE-F | ATGAGTTTTCAACTTCATCC | 850 bp | 56°C | [20] |
| | catE-R | TTAGGCTTCTCCTTTTATTTTGA | | | |

account for non-biological losses and validate the protocol. The volume of extracted oil was deducted from the previously weighed beaker.

The percentage of diesel oil degraded was calculated as follows:

$$Y = \frac{(W\ flask\ +\ initial\ oil - W\ flask\ +\ residual\ oil)}{(W\ flask\ +\ initial\ oil - W\ flask)}\ X\ 100\%$$

Here, Y = Percentage of oil degradation

### Characterization of residual crude oil

Fourier Transform Infrared Spectroscopy (FTIR) is a powerful analytical technique used to identify and quantify various chemical compounds in a sample based on their molecular vibrations [22]. A total of 3 samples (original diesel oil for control and residual oil 1,2 after degradation) were selected for FTIR analysis. Sample preparation and processing for FTIR analysis were made up following the guidelines of the Center for Advanced Research & Sciences, Dhaka University.

### Whole genome sequencing (WGS) of selected strain

Whole-genome sequencing of the target isolate was performed using the NextSeq 2000 Illumina platform at the Biotech Concern. The quality of the raw FASTQ files was assessed using FastQC (v0.12.1) [23], and low-quality bases and adapter sequences were trimmed using Trimmomatic (v0.39) [24] with parameters SLIDINGWINDOW:4:20, MINLEN:50, and ILLUMINACLIP for adapter removal. After trimming, the good-quality reads were subjected to de novo assembly using SPAdes v4.0.0 [25], and the assembled genome was taxonomically identified using the KmerFinder V3.2 tool [26]. To further confirm species identity and assess genomic similarity, FastANI [27] was used to calculate the Average Nucleotide Identity (ANI) between the target isolate and reference genomes from the NCBI database. Multilocus sequence typing (MLST) was performed using the MLST tool from the Center for Genomic Epidemiology (CGE) [28] which identified sequence types (STs) based on conserved housekeeping genes. A graphical map of the circular genome was generated using the Proksee server [29]. The assembled sequence undergone annotation through Prokka [30], RAST [31] and Prokaryotic genome annotation pipeline (PGAP) of NCBI [32]. Antibiotic resistance genes, virulence factors, and plasmid-associated genes were identified using ABRicate (v1.0.1) [33] with the ANNOT [34], NCBI AMR [35], CARD (v3.2.4) [36], VFDB (2023 release) [37] and PlasmidFinder databases [38]. The biosynthetic gene clusters were determined through antiSMASH7.0 tool [39]. Analysis of different subsystems and functional metabolic pathways was carried out through the Patric server [40]. Pathogenicity profiling and prediction of the pathogenicity of all the isolates towards human hosts was carried out through PathogenFinder (v1.1) [41] webtool of Center for Genomic Epidemiology. The *alkB* protein is essential for the oil degradation pathway [42]. Therefore, we conducted mutation analysis to identify variations within the *alkB* gene of *Acinetobacter sp*.

### Phenotypic characterization of antibiotic resistance

Antibiotic susceptibility testing (AST) was carried out on the isolates by using Mueller-Hinton agar by the Kirby-Bauer disk diffusion method [43]. The results were categorized as sensitive, intermediate, or resistant according to the Clinical and Laboratory Standards Institute's (CLSI) M100-Ed35:2024 guidelines. The following antibiotic discs were used in the assay: ampicillin, ceftazidime, cefotaxime, cefepime, imipenem, meropenem, gentamycin, tobramycin, kanamycin, ciprofloxacin, levofloxacin, tetracycline, doxycycline, azithromycin, and chloramphenicol.

## Results and discussion

### Isolation and identification of potential oil-degrading bacteria

Oil spills pose a significant environmental threat globally, with Bangladesh being particularly vulnerable. Frequent spills occur at Chittagong's port-centric shipyards and fuel stations, often due to negligence or inadequate safety practices. The Sundarbans,

the world's largest mangrove forest, suffered severe ecological damage from a major oil spill in recent years [44,45]. In Dhaka, routine leaks at fuel stations and industrial discharge—including oils and toxic chemicals—have led to critical water pollution. Studies reveal oil contaminants in 65% of water samples collected near industrial areas [46]. Hence, from Bangladesh's stand-point, spillage of crude oil onto soil and its dispersal into the environment is inevitable, and thus we need to act with a state-of-the-art bioremediation approach specifically focusing on the genomic potential of the indigenous isolates. In this study, we identified a wide range of bacteria from oil-contaminated sites of Dhaka. A total of 34 morphologically distinct bacterial isolates were recovered from diesel-enriched Bushnell Haas medium inoculated with soil samples collected from four filling stations in Dhaka. Following incubation and purification on diesel-supplemented agar plates, these isolates demonstrated visible growth under selective conditions, suggesting potential diesel-degrading capability. 16S rRNA sequencing identified the bacterial com-position as follows: *Acinetobacter baumannii* was the most prevalent species, with 14 isolates (41.17%), followed by *Pseudo-monas otitidis* and *Klebsiella pneumoniae*, each with 4 isolates (11.76%) (Table 2). *Pseudomonas aeruginosa* was represented by 3 isolates (8.82%), while *Pseudomonas nitroreducens* and *Capriavidus gilardi* each had 2 isolates (5.88%). *Pseudomonas* and *Acinetobacter* species are consistently effective in oil degradation across studies, though efficiency varies with environ-mental conditions and microbial partnerships (e.g., co-cultures achieved higher degradation rates) [47]. Additionally, *Cellulo-monas hominis*, *Stenotrophomonas maltophilia*, *Glutamicibacter urotoxydans*, *Pseudomonas mendocina*, and *Pseudomonas taiwanensis* were each represented by 1 isolate (3%). The qualitative biofilm formation assay using CRA plates revealed that only a subset of isolates formed black, dry, crystalline colonies indicative of strong biofilm production. These included *Capriavi-dus gilardi* (MB 756), *A. baumannii* (MB 503, MB 504, and MB 505), *P. otitidis* (MB 257 and MB 253), *P. aeruginosa* (MB 752), *Klebsiella pneumoniae* (ML 025), and *Pseudomonas mendocina* (ML 075). The quantitative microtiter plate assay revealed that among the 34 isolates, 3 were strong biofilm-formers, 12 were moderate, 9 were weak, and the remaining 10 showed no detectable biofilm formation. The strong biofilm-formers were from *A. baumannii* (MB 503 and MB 505) and *P. aeruginosa* (MB 752). The ability to form biofilms underpins the ecological fitness of these strains in hydrocarbon-rich environments, facilitating adherence to hydrophobic substrates and enhancing pollutant breakdown efficiency.

These isolates were screened for the presence of *alkB* and *catE* genes. Out of 34 isolates, 32.35% (11 isolates) were positive for *alkB*, whereas 58.82% (20 isolates) tested positive for *catE*, indicating a higher prevalence of catechol-2,3-dioxygenase producers compared to alkane hydroxylase producers. The *alkB* gene was detected in *K. pneumoniae* (ML 50, ML 100), *P. mendocina* (ML 075), *P. nitroreducens* (MB 502 and MB 1002), *A. baumannii* (MB 503, MB 505, MB 751,

**Table 2. Bacterial composition identified by 16S rRNA sequencing.**

| Identified Microorganisms | Designated numbering | Count |
|---|---|---|
| *Pseudomonas otidis* | MB 251, MB 253, MB 257, MB 1005 | 4 |
| *Pseudomonas aeruginosa* | MB 254, MB 752, MB 252 | 3 |
| *Klebsiella pneumoniae* | MB 250, ML 25, ML 50, ML 100 | 4 |
| *Acinetobacter baumannii* | MB 255, MB 256, MB 501, MB 503, MB 504, MB 505, MB 506, MB 751, MB 753, MB 754, MB 755, MB 757, MB 1001, MB 1003 | 14 |
| *Pseudomonas nitroreducens* | MB 502, MB 1002 | 2 |
| *Cellulomonas hominis* | MB 507 | 1 |
| *Capriavidus gilardi* | MB 756, MB 1006 | 2 |
| *Strenotrophomonas maltophila* | MB 1007 | 1 |
| *Glutamicibacter urotoxydans* | MB 1004 | 1 |
| *Pseudomonas mendocina* | ML 075 | 1 |
| *Pseudomonas taiwanensis* | ML 125 | 1 |

MB 755, MB 1001, MB 1003). The *catE* gene was detected in *P. otitidis* (MB 253), *P. aeruginosa* (MB 252, MB 254, MB 752), *A. baumannii* (MB 255, MB 256, MB 501, MB 503, MB 504, MB 505, MB 506, MB 751, MB 754, MB 755, MB 1000, MB 1001, MB 1003), *P. nitroreducens* (MB 750), *C. hominis* (MB 507), and *C. gilardi* (MB 500). The prevalence of *catE* in these isolates aligns with global findings that catechol dioxygenases are critical for aromatic hydrocarbon breakdown [48,49]. Previous studies have shown that *alkB*-encoded alkane hydroxylases are widely used as biomarkers for alkane degradation in marine and soil environments [50,51], while *catE*-encoded catechol-2,3-dioxygenases play a central role in aromatic compound degradation and have been applied in the bioremediation of oil-contaminated soils and groundwater [52,53]. All 34 isolates were then evaluated for degrading ability via turbidimetric analysis, and 11 isolates that exhibited higher growth rates were selected for further assessment using gravimetric analysis. Among these, isolate MB 1002 (*P. nitroreducens*) demonstrated the highest degradative capability at 46.92%, followed by MB 751 (*Acinetobacter baumannii*) at 41.18% and MB 750 (*Pseudomonas aeruginosa*) at 39.16% (Fig 1). Previous studies have reported variable diesel degradation efficiencies under differing experimental conditions, as summarized in Table 3. For instance, the diesel-degrading efficiency of *A. baumannii* has been shown to reach up to 99% at 37 °C in controlled laboratory settings [54]. In contrast, our isolates demonstrated substantial degradation efficiency at 30 °C, which more closely reflects typical environmental conditions, underscoring their potential applicability in natural or field-based bioremediation efforts.

The MB 1002 and MB 751 both were positive for *alkB*. This suggests a potential role of *alkB* in the oil-degrading ability of these isolates. The biodegradation observed by MB 1002 and MB 751 were subjected to Fourier-transform infrared spectroscopy (FTIR) analysis.

## Fourier-transform infrared spectroscopy (FTIR) analysis

The FTIR analysis of diesel oil and its degraded forms by bacterial strains MB 1002 and MB 751 reveals significant structural changes in hydrocarbon composition, indicating biodegradation (Fig 2). The original diesel oil exhibits strong peaks at 732.95 cm$^{-1}$, 1458.18 cm$^{-1}$, 2858.51 cm$^{-1}$, and 2922.16 cm$^{-1}$, corresponding to methylene (-CH$_2$-) and methyl (-CH$_3$) groups (Table 4). These peaks are still present in the residual oils (MB 1002 and MB 751), but their intensities have diminished, suggesting partial degradation. Additionally, MB 1002 and MB 751 show new peaks around 881.47 cm$^{-1}$, 1043.49 cm$^{-1}$, and 1130.29 cm$^{-1}$, indicating the presence of aromatic C-H and alkene groups. The detection of aromatic C-H (881 cm$^{-1}$) aligns with India's FTIR data [58]. Notably, MB 1002 showed new hydroxyl group formation (3373.5 cm$^{-1}$), suggesting alcohol production similar to *Mariobacter spp.* Pathways [59]. Meanwhile, MB 751 developed a distinct carboxyl peak at 1712.79

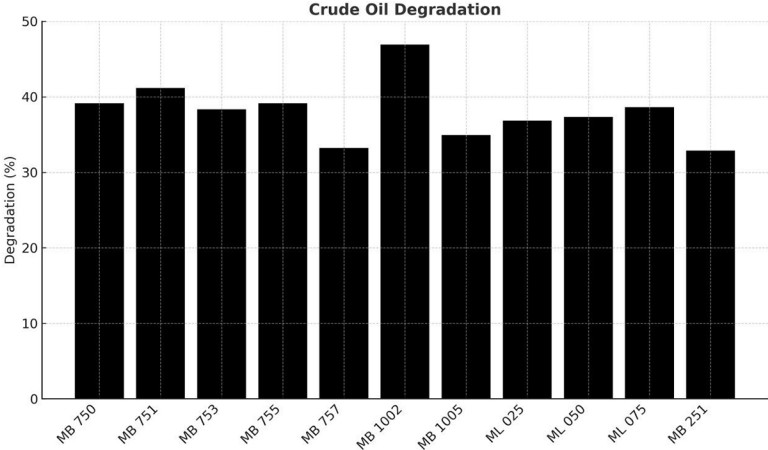

**Fig 1. The percentage of diesel oil degradation.** *P. nitroreducens* MB1002 showed the highest degradation efficiency (46.92%), followed by *A. baumannii* MB751 (41.18%) and *P. aeruginosa* MB750 (39.16%).

**Table 3. Comparative table to present the diesel degradation efficiency.**

| Organisms | Type | Diesel Degradation Efficiency | Notes | Source |
|---|---|---|---|---|
| *Acinetobacter baumannii* DUEMBL6 | Bacterial strain | ≈ 41% | ~30 °C, neutral pH, and 7–15 days incubation | This Study |
| *Pseudomonas aeruginosa* | Bacterial strain | 39.16% | ~30 °C, neutral pH, and 7–15 days incubation | This Study |
| *Acinetobacter baumannii* | Bacterial strain | >99% | 5 days incubation; pH 7; 35–37°C | [54] |
| *Arthrobacter spp.* Antarctic strains AQ5–05 and AQ5–06 | Bacterial strains | 34.5% to 47.5% | Cold-adapted; 10–15°C optimal temp | [55] |
| *Vibrio sp.* ZL2, *Acinetobacter sp.* ZY3, *Enterobacter cloacae* GM6 | Bacterial consortium | 89.66% to 93.65% | Day 3 degradation; optimized inoculation ratios | [56] |
| *Aspergillus ustus* and *A. alternata* | Fungal consortium | Up to 97.73% | Soil and water media; 168 h incubation | [57] |

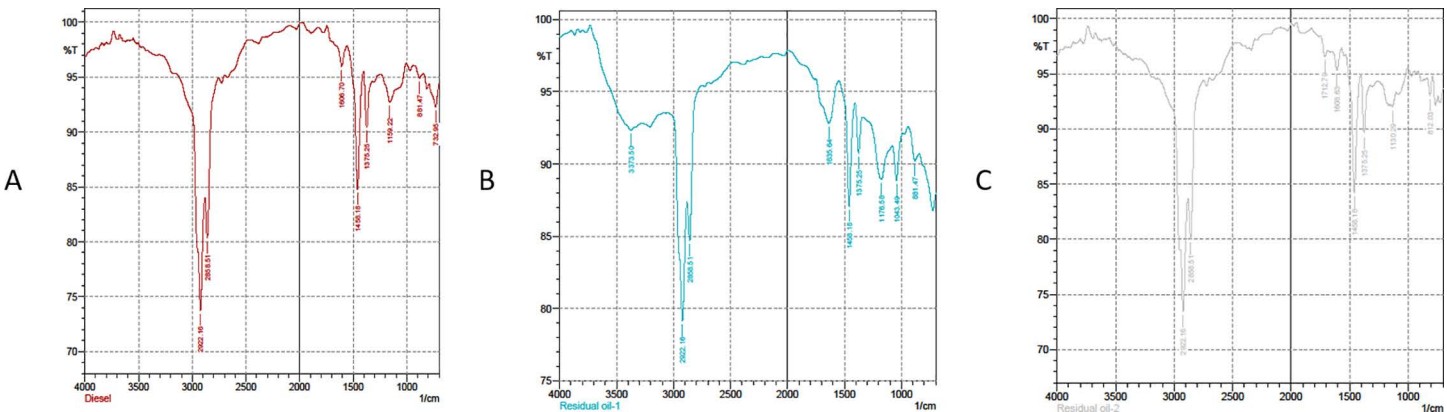

**Fig 2. Fourier-transform infrared (FTIR) spectra of diesel and its degraded forms.** Spectra show (A) unmodified diesel, (B) residual oil after degradation by *P. nitroreducens* MB1002, and (C) residual oil after degradation by *A. baumannii* MB751. Reduced intensities of hydrocarbon peaks (732–2922 cm⁻¹) indicate partial degradation, while new bands (881–1130 cm⁻¹) reflect aromatic and alkene groups. Hydroxyl (3373 cm⁻¹) and carboxyl (1712 cm⁻¹) peaks suggest oxidative transformation of hydrocarbons, with stronger oxidation in MB751.

cm⁻¹, indicative of ketone, aldehyde, or carboxylic acid formation, consistent with *Marinobacter*-mediated oxidative degradation [49]. The appearance of these oxygen-containing functional groups, absent in the original diesel, confirms that microbial activity has altered the chemical structure of diesel hydrocarbons. The results suggest that both MB 1002 and MB 751 contribute to diesel degradation, with MB 751 showing stronger oxidation patterns due to the presence of carboxyl functional groups. Comparative analysis revealed that *P. nitroreducens* (MB 1002) outperforms cold-adapted *Oleispira* (32% PAH degradation) [59] but lags behind *Sphingomonas* consortia [60]. Nonetheless, the degradation efficiency of these strains remains moderate compared to *Bacillus spp.* from China (66%) [61] and optimized consortia from India (95%) [49].

## Whole genome sequence analysis, annotation and identification

Among the tested strains, MB 751 (*A. baumannii*) was selected for whole-genome sequencing (WGS) based on its superior and more complete degradation efficiency. Sequence-based identification through KmerFinder further confirmed the selected isolate to be *A. baumannii*. Multilocus sequence typing (MLST) analysis revealed an unknown sequence type, suggesting that the isolate represents a novel strain of *A. baumannii.* Afterwards, the isolate was designated as "*Acinetobacter baumannii* DUEMBL6". Further analysis revealed a 97.93% Average Nucleotide Identity (ANI) with another *A. baumannii* strain (GCA_016746035.1),

**Table 4. FTIR Analysis Data.**

| Original Diesel oil | | MB 1002 | | MB 751 | |
|---|---|---|---|---|---|
| Wavelength | Functional groups | Wavelength | Functional groups | Wavelength | Functional groups |
| 732.95 | Methylene (>CH2) Methyl (>C-H) asym./sym. Stretch,) | 881.47 | C-H1,3- Disubstitution (meta) | 812.03 | Aromatic C-H out-of- plane bend |
| 881.47 | C-H1,3- Disubstitution (meta) | 1043.49 | Methylene (>CH2) Cyclohexane ring vibrations, | 1130.29 | Aromatic C-H out-of- plane bend |
| 1159.22 | Methyne (>CH−), Aromatic C- H in-plane bend | 1176.58 | Methyne (>CH−), | 1375.25 | Methyl (−CH3): gem- Dimethyl or"iso"- (doublet) |
| 1375.25 | Methyl (−CH3): gem- Dimethyl or "iso"- (doublet) | 1375.25 | Methyl (−CH3): gem- Dimethyl or "iso"- (doublet) | 1458.18 | Methylene (>CH2) Methylene C-H bend |
| 1458.18 | Methylene (>CH2) Methylene C-H bend | 1458.18 | Methylene (>CH2) Methylene C-H bend | 1608.63 | Aromatic ring (aryl) C = C-C Aromatic ring stretch |
| 1606.7 | C = C-C Aromatic ring stretch | 1635.64 | Olefinic (alkene)Alke- nyl C = C stretch | 1712.79 | Aromatic Combination bands, Carboxylic acid, Ketone, Aldehyde |
| 2858.51 | Methylene (>CH2) Methylene C-H asym./sym. Stretch | 2858.51 | Methylene (>CH2) Methylene C-H asym./sym. Stretch | 2858.51 | Methylene (>CH2) Methylene C-H asym./sym. Stretch |

which provides strong evidence for the accurate identification of the isolate. The assembled whole genome sequence was deposited in GenBank of NCBI under the accession number JBLODW000000000. The analysis of the genome indicated good quality of the genome considering the completeness (97.33%) and low contamination (0.83%). Additionally, the genome size and GC content of the sequenced isolates aligned well with the established genomes of *A. baumannii*. Following the process of de novo assembly and annotation, the CDS ratio of 0.946 provided evidence of a well-executed and successful assembly. This conclusion is based on the fact that the standard CDS ratio endorsed by NCBI typically falls between 0.8 and 1.2 [62]. The genome annotation and mapping revealed significant features of the genome which are listed in (Table 5 and Fig 3).

### Prevalence of hydrocarbon degrading genes

Functional annotation of *A. baumannii* DUEMBL6 revealed the presence of multiple genes associated with hydrocarbon degradation, including alkane 1-monooxygenase (*alkB*), which catalyzes the initial oxidation of alkanes [42]; alkanesulfonate monooxygenase (*ssuD*), involved in the degradation of sulfonated alkanes [63]; catechol O-methyltransferase (*COMT*), which plays a role in the metabolism of aromatic compounds [64]; and catechol 1,2-dioxygenase, which cleaves catechol rings during aromatic hydrocarbon degradation [65]. These genes are known to play crucial roles in the breakdown of both aliphatic and aromatic hydrocarbons, highlighting the strain's potential for bioremediation. Mutation analysis of the *alkB* protein in *A. baumannii* DUEMBL6 revealed two specific mutations, I138F and A334T, when compared to other closely related strains. These mutations may indicate a variant of *alkB* with enhanced metabolic activity, potentially leading to more efficient degradation of hydrocarbons [58].

### Identification of biosynthetic gene clusters

The genomic analysis of this diesel-degrading bacterium using antiSMASH revealed a diverse array of secondary metabolite biosynthetic gene clusters (BGCs), highlighting their potential ecological and biotechnological significance. A total of 7 secondary metabolite BGCs were identified under relaxed detection settings, spanning multiple functional categories

**Table 5. General features of the assembled whole genome.**

| PGAP Annotation | | Pokka Annotation | |
| --- | --- | --- | --- |
| contigs | 293 | contigs | 293 |
| Genome size (bases) | 4Mb | bases | 3994935 |
| Genes | 3940 | CDS | 3735 |
| Protein-coding | 3769 | rRNA | 10 |
| rRNA | 14 | Repeat_regions | 3 |
| tRNA | 64 | tRNA | 64 |
| CRISPR Arrays | 2 | tmRNA | 1 |
| Other features | | | |
| Genome Coverage | 54.0X | GC percentage | 39 |

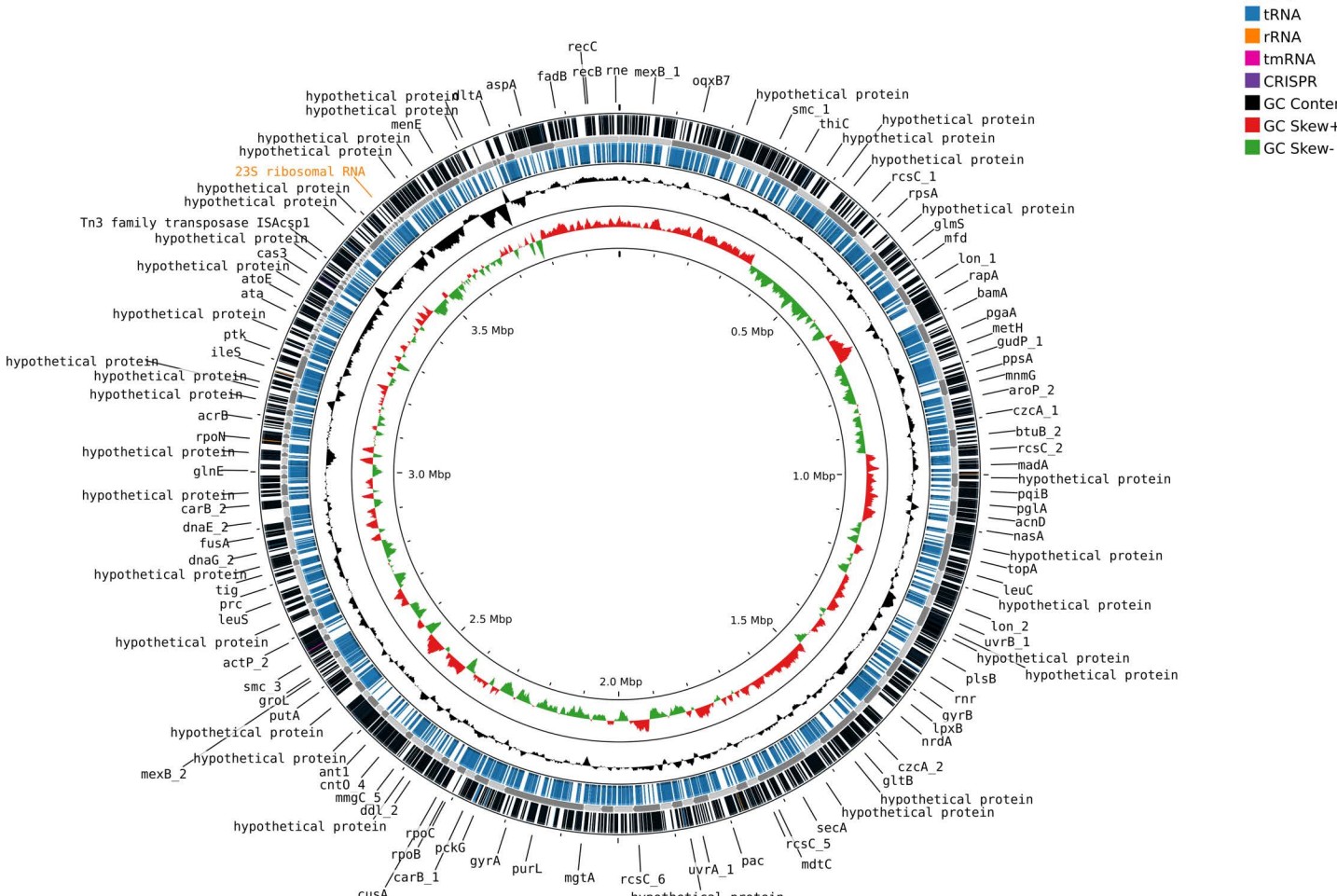

**Fig 3. Whole genome mapping of *A. baumannii* DUEMBL6.** Circular representation of the *A. baumannii* DUEMBL6 genome showing coding sequences (CDS), GC content, and GC skew.

(Fig 4). Among these, the NRP-metallophore biosynthetic cluster (Region 23.1) exhibited 83% similarity to the well-characterized acinetobactin cluster, a known siderophore involved in iron acquisition [66]. Additionally, a beta lactone-producing cluster (Region 39.1) showed 20% similarity to the mycosubtilin cluster, which is associated with antimicrobial activity [67]. Furthermore, arylpolyene biosynthetic clusters (Regions 79.1 and 211.1) were detected, with potential roles in oxidative stress resistance and biofilm formation [68]. Notably, Region 81.1 (NI-siderophore cluster) displayed 100% similarity to the Baumannoferrin A/B biosynthetic pathway, suggesting a significant contribution to metal ion sequestration and environmental adaptation [69].

The identification of these BGCs in a diesel-degrading bacterium suggests that secondary metabolites may play a crucial role in the organism's survival and competitiveness in hydrocarbon-rich environments [70]. For instance, the

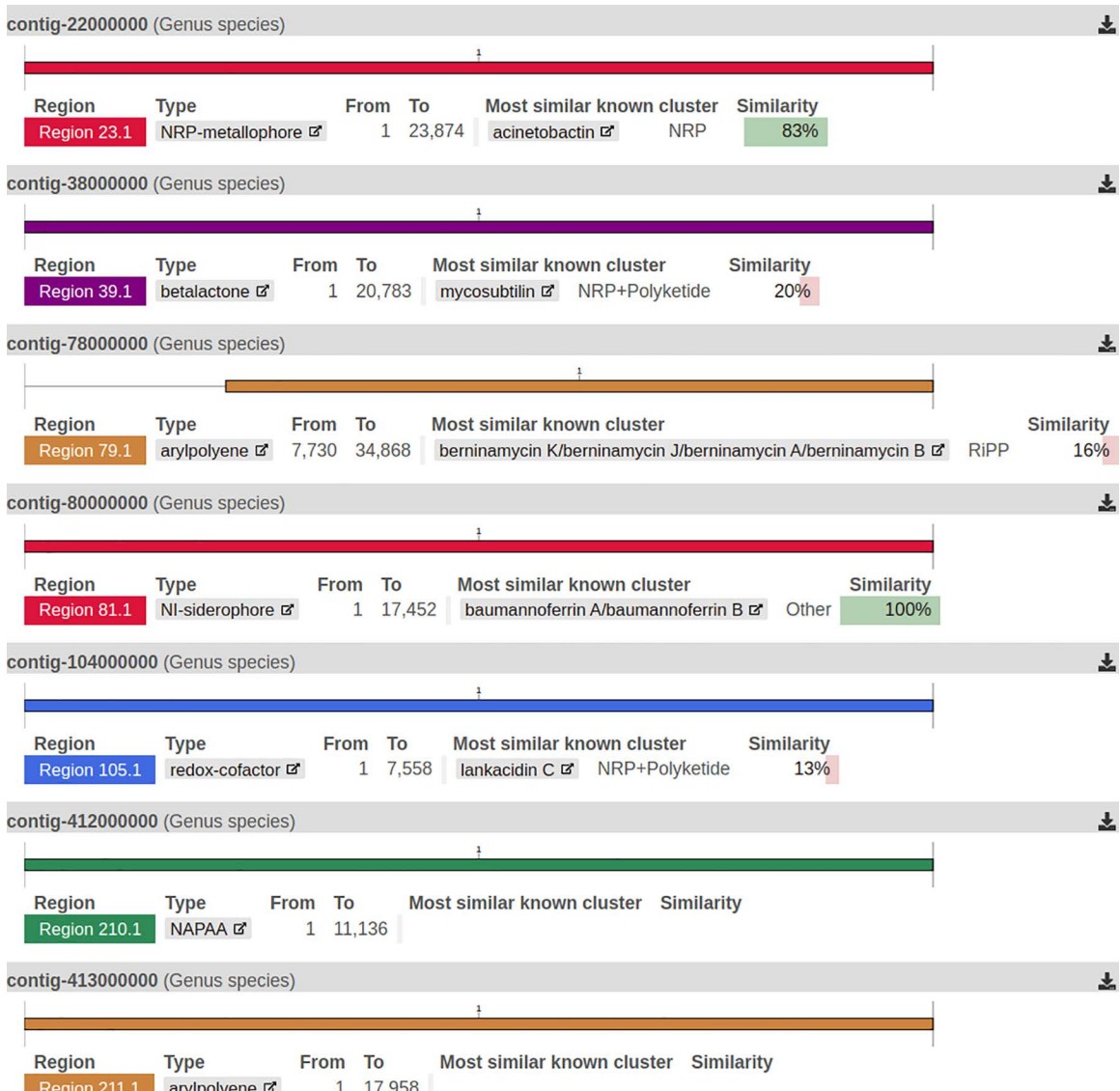

**Fig 4. Biosynthetic gene clusters of *A. baumannii* DUEMBL6.** Genome mining using antiSMASH revealed seven secondary metabolite biosynthetic gene clusters, including NRP-metallophore, beta-lactone, arylpolyene, and NI-siderophore clusters.

production of siderophores (e.g., NI-siderophore) could enhance the bacterium's ability to acquire iron, a limiting nutrient in many environments, while the production of antimicrobial compounds (e.g., betalactone) could provide a competitive advantage against other microorganisms [71]. These findings underscore the ecological and biotechnological potential of *A. baumannii* DUEMBL6, not only as a hydrocarbon degrader but also as a producer of bioactive secondary metabolites that could be harnessed for industrial and environmental applications.

## Pathway and subsystem analysis

The analysis of metabolic pathways revealed thirteen distinct classes involved in a range of essential biological processes, including carbohydrate metabolism, amino acid metabolism, glycan biosynthesis and metabolism, signal transduction, lipid metabolism, energy metabolism, biosynthesis of polyketides, secondary metabolite biosynthesis, and xenobiotics biodegradation and metabolism (Fig 5). The identified pathway class xenobiotics degradation and metabolism is particularly important in biotechnology due to its role in breaking down environmental pollutants and synthetic compounds [72,73]. We identified pathways for degrading diverse xenobiotics, including 1,4-dichlorobenzene degradation, 1- and 2-methylnaphthalene degradation, 2,4-dichlorobenzoate degradation, atrazine degradation, benzoate degradation via hydroxylation, bisphenol A degradation, caprolactam degradation, DDT degradation, drug metabolism (cytochrome P450 and other enzymes), ethylbenzene degradation, fluorobenzoate degradation, geraniol degradation, metabolism of xenobiotics by cytochrome P450, naphthalene and anthracene degradation, styrene degradation, tetrachloroethene degradation, toluene and xylene degradation, trinitrotoluene degradation, and γ-hexachlorocyclohexane degradation. Each of these pathways comprises multiple enzymes that contribute to the degradation of diesel components, including both aromatic hydrocarbons and aliphatic compounds (Fig 6). Catechol 1,2-dioxygenase and protocatechuate 3,4-dioxygenase are critical for cleaving aromatic rings in polycyclic aromatic hydrocarbons (PAHs) like naphthalene and phenanthrene, converting them to tricarboxylic acid (TCA) cycle intermediates [74,75]. Benzoate 1,2-dioxygenase initiates the breakdown of monoaromatic compounds such as toluene and xylene, common diesel constituents [76,77]. Enzymes like 3-hydroxyacyl-CoA dehydrogenase and enoyl-CoA hydratase participate in β-oxidation pathways, degrading straight-chain alkanes into acetyl-CoA [78]. Cyclohexanone monooxygenase facilitates the degradation of cyclic alkanes, while benzaldehyde dehydrogenase and alcohol dehydrogenase metabolize intermediate aldehydes and alcohols generated during alkane oxidation [79,80]. These pathways align with established mechanisms for microbial diesel degradation, as demonstrated in *Pseudomonas* and *Acinetobacter* species adapted to hydrocarbon-rich environments [81,82].

We also analyzed the subsystems of *A. baumannii* DUEMBL6, which were categorized into eleven superclasses and twenty-six functional classes (Fig 7). These subsystems are involved in essential biological processes, including metabolism, protein processing, stress response and defense, virulence, maintenance of cell envelope integrity, respiration, membrane transport, cellular processes, and regulation and cell signaling. Further classification revealed a total of seventy-seven distinct subclasses. This subsystem analysis highlights that *A. baumannii* DUEMBL6 harbors a diverse array of genes contributing to multiple functional categories critical for survival, adaptation, and potential pathogenicity.

## Antibiotic resistance genes and mobile genetic elements

The presence of antibiotic resistance genes in isolates showing bio remediation ability is not desirable since they can serve as a source of resistance for other pathogens. To assess this risk, we investigated the antibiotic resistance genes in the isolate by screening multiple databases, including ResFinder, NCBI AMR, MEGARes, CARD, and ARG-ANNOT, using ABRicate. The results suggest the bacteria as a multi-drug resistant isolate with the presence of 26 resistance genes (S1 Table). The β-lactamase genes detected included *blaOXA-338*, *blaOXA-781*, *blaOXA-800*, *blaADC-25*, *blaADC-2*, *blaADC-158*, and *blaA1*, which contribute to resistance against β-lactam antibiotics, including cephalosporins and carbapenems. Among them, *blaOXA*-type genes are associated with carbapenem hydrolysis [83], while *blaADC* genes encode class C β-lactamases that can degrade cephalosporins [83,84]. Phenotypically, the isolate displayed resistance to ampicillin,

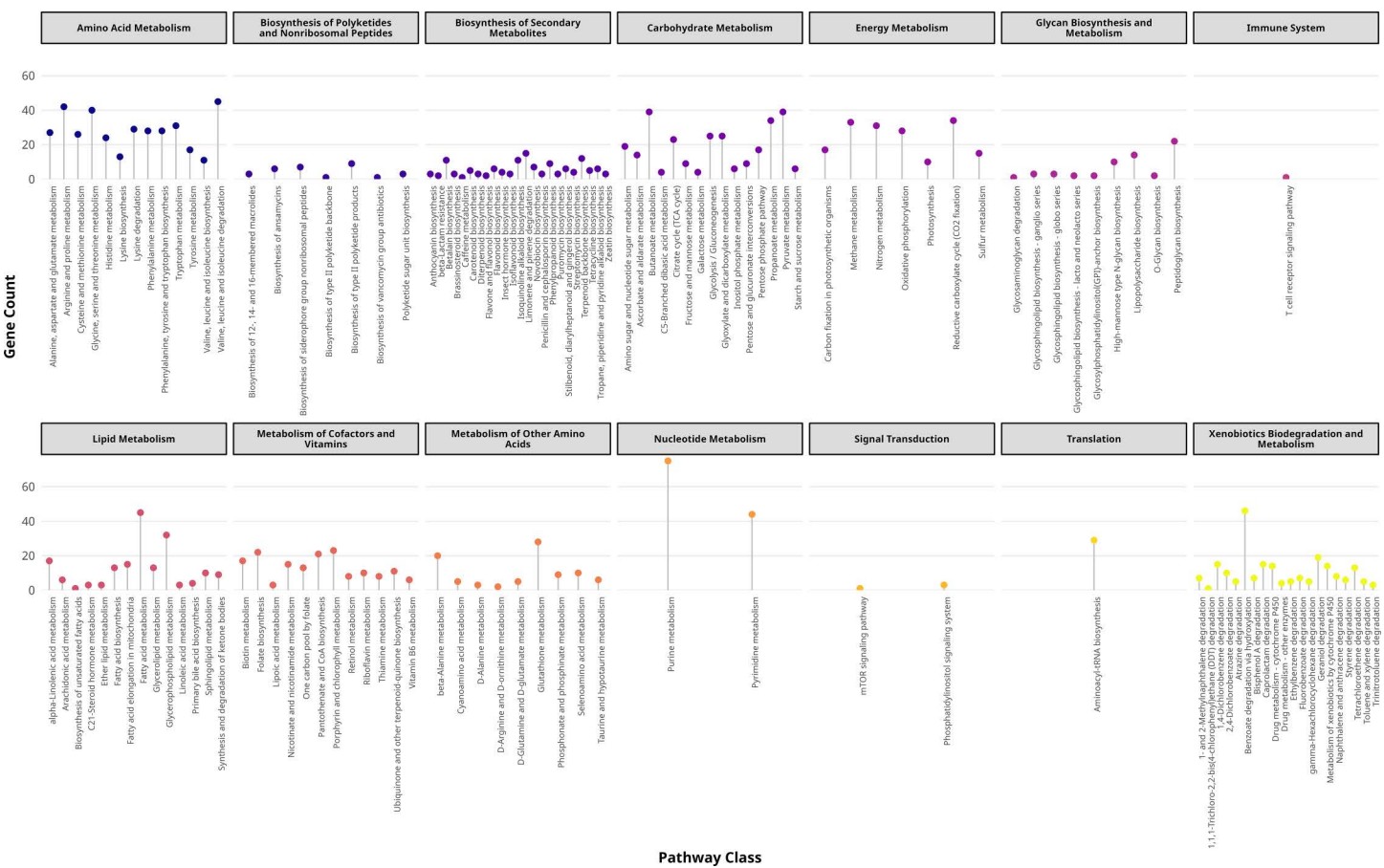

**Fig 5. Pathway analysis of _A. baumannii_ DUEMBL6.** Metabolic pathway analysis revealed thirteen major classes involved in core cellular and environmental functions, including carbohydrate, amino acid, lipid, and energy metabolism, as well as secondary metabolite and xenobiotic degradation pathways.

ceftazidime, cefotaxime, cefepime, imipenem, and meropenem (Table 6), consistent with the genomic detection of OXA and ADC enzymes. Aminoglycoside resistance was conferred by _ant(3'')-IIa_, which encodes an aminoglycoside nucleotidyltransferase enzyme, modifying aminoglycosides such as gentamicin, tobramycin, and kanamycin, reducing their efficacy [85,86]. This correlated well with the phenotypic resistance observed against gentamicin, tobramycin, and kanamycin in our susceptibility assays. A review highlights that ARGs, particularly β-lactamases, are abundant in petroleum-contaminated soils due to co-selection pressures from hydrocarbons and metals, which favor resistant strains [87].

Multiple efflux pump-related genes were also detected, including _adeT1_, _adeT2_, _adeS_, _adeR_, _adeA_, _adeB_, _abeS_, _abeM_, _adeK_, _adeJ_, _adeI_, _adeH_, _adeG_, _adeF_, and _adeL_. The AdeABC efflux system (_adeA_, _adeB_, and _adeC_) is known for mediating resistance to fluoroquinolones, aminoglycosides, and tigecycline, while AbeM and AbeS contribute to multidrug resistance in _A. baumannii_ [88,89]. Regulatory elements _adeR_ and _adeS_ were also present, suggesting potential overexpression of efflux pumps [90,91]. Consistently, the strain exhibited phenotypic resistance to ciprofloxacin, levofloxacin, doxycycline and tetracycline, supporting the functional relevance of these efflux systems. Additionally, MexT, a transcriptional regulator, was identified, which is known to upregulate the MexEF-OprN efflux pump in _Pseudomonas aeruginosa_, leading to resistance against fluoroquinolones and chloramphenicol [92]. The presence of _abaQ_ suggests a role in additional efflux mechanisms, potentially affecting aminoglycoside resistance [88,93]. The _amvA_ gene, encoding a

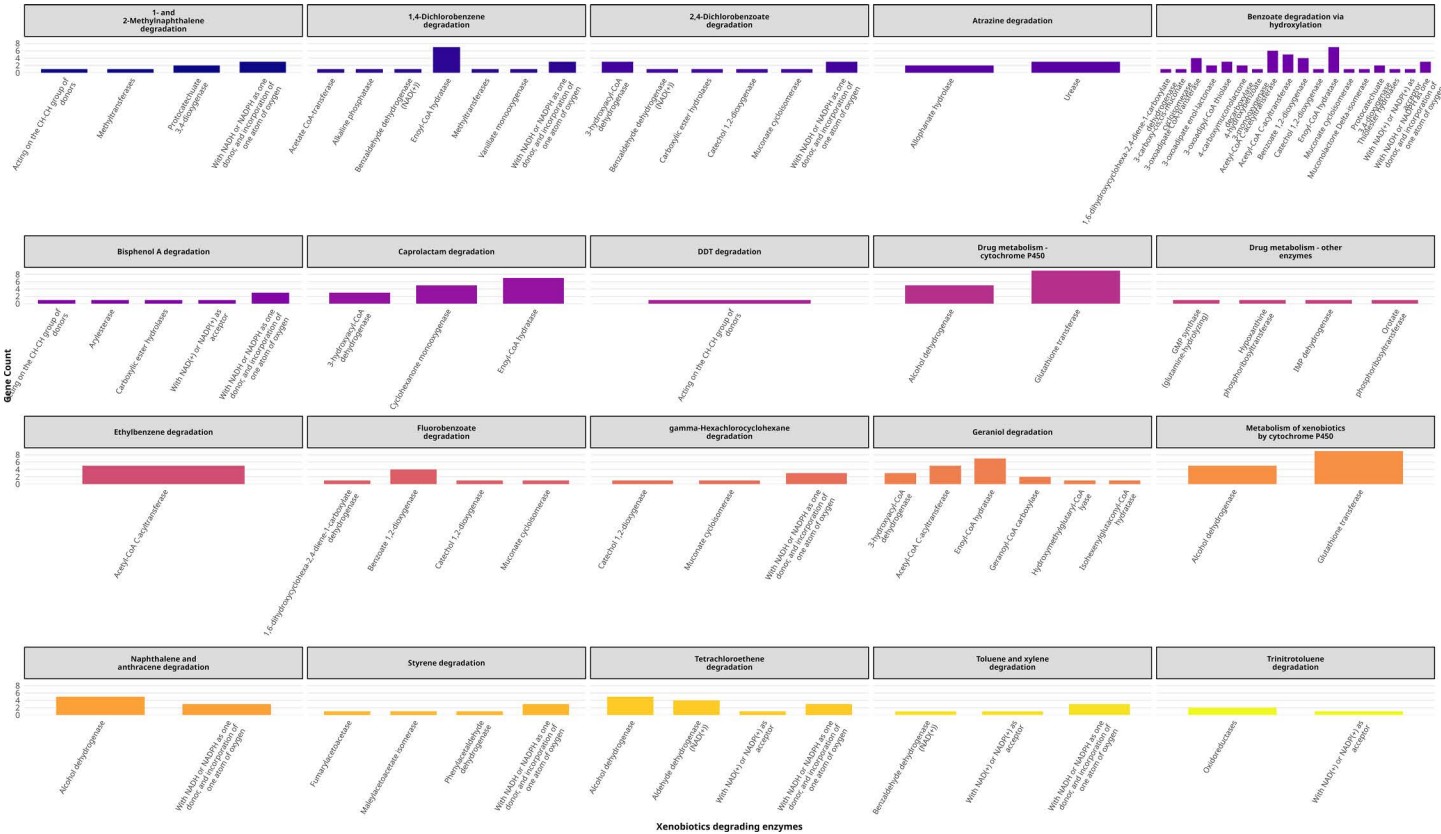

**Fig 6. Pathway classes for xenobiotics degradation and metabolism.** Identified xenobiotic degradation pathways include 1,4-dichlorobenzene, toluene, xylene, naphthalene, and bisphenol A degradation, among others. These pathways involve enzymes such as catechol 1,2-dioxygenase and benzoate 1,2-dioxygenase, highlighting the strain's potential for biodegradation of aromatic and aliphatic hydrocarbons.

major facilitator superfamily (MFS) efflux pump, was also detected, further supporting the isolate's ability to expel various antibiotics, including macrolides [88,93]. Phenotypic resistance to azithromycin and chloramphenicol further reflected the genomic predictions for these efflux mechanisms.

The combination of β-lactamase production, aminoglycoside-modifying enzymes, and multiple efflux pumps suggests a robust antibiotic resistance mechanism in the isolate. However, no plasmid-associated genes were identified after screening the PlasmidFinder database using ABRicate, indicating that the organism likely has a limited ability to spread resistance genes horizontally. This finding suggests that while the isolate possesses intrinsic resistance mechanisms, the absence of plasmid-mediated resistance reduces the risk of widespread dissemination of these genes to other pathogens in the environment.

### Virulence factor, metal resistance genes, and pathogenicity

The presence of virulence factors indicates both the bioremediation potential and pathogenic risk. In *A. baumannii* DUEMBL6, a total of 33 virulence genes were identified, as listed in S2 Table. The *csuA/B, csuA, csuB, csuC, csuD,* and *csuE* genes, which encode the chaperone-usher pili assembly system, were detected, highlighting their potential role in surface adherence and biofilm formation [94,95]. The presence of *pgaA*, *pgaB*, *pgaC*, and *pgaD* suggests that the bacterium has the ability to produce poly-β-(1,6)-N-acetylglucosamine (PNAG), a key biofilm matrix component that enhances environmental persistence and antibiotic resistance [94,95]. The regulatory genes *bfmR* and *bfmS*, which control biofilm

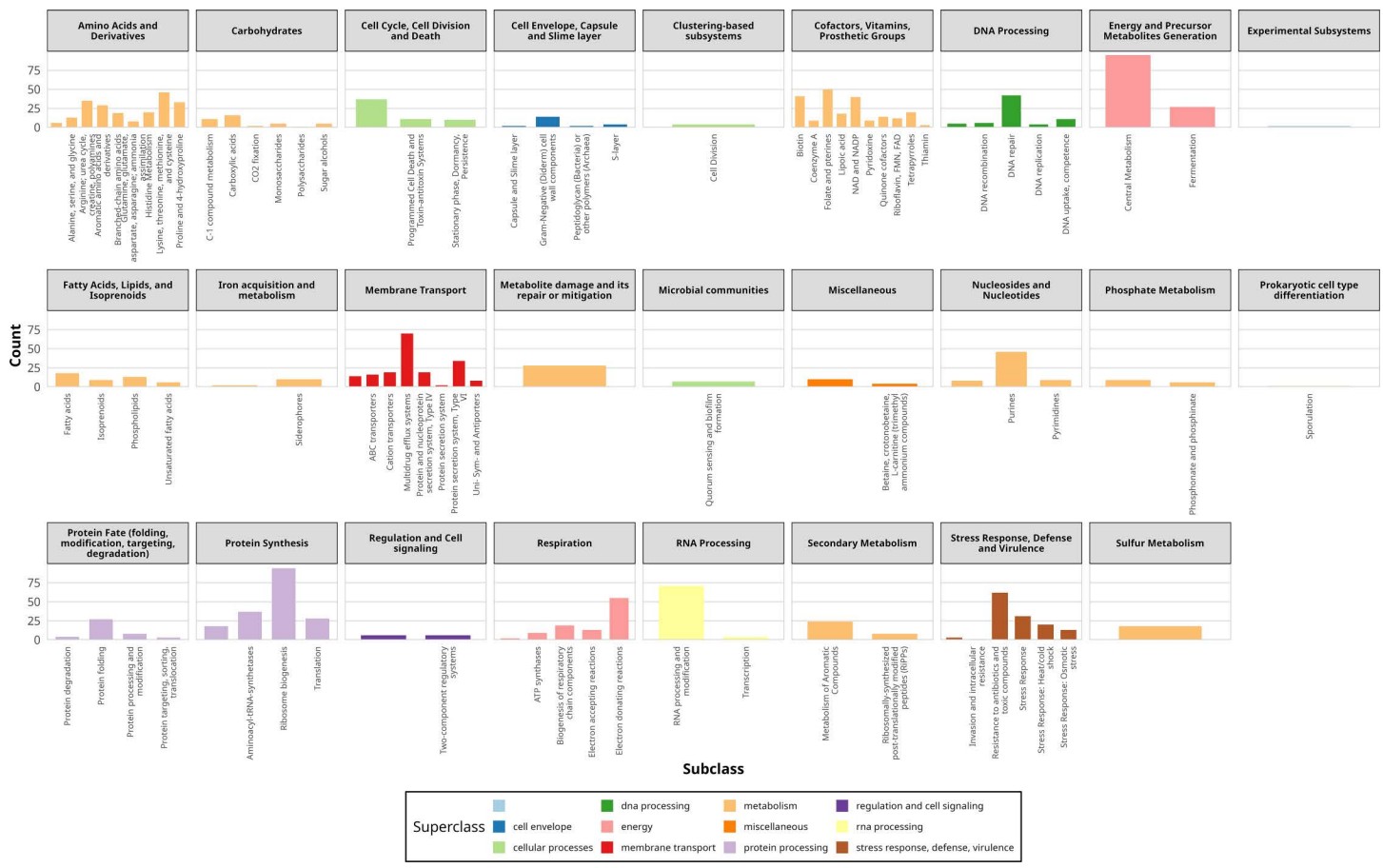

**Fig 7. Subsystem analysis of _A. baumannii_ DUEMBL6.** Subsystem annotation revealed eleven superclasses and twenty-six functional classes encompassing metabolism, stress response, virulence, membrane transport, and regulatory processes. These findings illustrate the genetic versatility and adaptive capacity of _A. baumannii_ DUEMBL6.

maturation, were also identified, suggesting a robust biofilm-forming capability [94,95]. The presence of phospholipase genes _plc_ and _plcD_ indicates the potential for membrane disruption and tissue invasion, contributing to the pathogenicity of the strain [86,87]. Iron acquisition systems were well-represented, with _bauA, bauB, bauC, bauD, bauE, bauF, basA, basB, basC, basD, basF, basG, basH, basJ, barA,_ and _barB_ detected. These genes encode proteins involved in the biosynthesis and transport of acinetobactin and other siderophores, which are critical for survival in iron-limited environments such as the human host [94,95]. The presence of _entE_ further supports the strain's ability to sequester iron, enhancing its virulence. Outer membrane protein (_ompA_) and regulatory genes such as _abaR_ were also detected, which are known to play roles in immune evasion and antimicrobial resistance [95,96].

Along with virulence and antimicrobial resistance genes, several metal resistance genes were also detected in the genome of _A. baumannii_ DUEMBL6, suggesting adaptation to metal-contaminated environments. _ZitB_ and _YeiR_ encode zinc efflux transporters, regulating intracellular zinc levels to prevent toxicity [97]. _CzcA_ and _CzcB_ form a cobalt-zinc-cadmium efflux system, crucial for heavy metal resistance [97]. The _FtsH_ zinc-dependent protease aids in stress response by degrading misfolded proteins [93]. _mntP_ regulates manganese efflux, balancing metal homeostasis [98]. _KefF_, _KefC_, and _KdpC_ contribute to potassium transport, protecting against acidic stress and osmotic imbalance [99]. _FdhD_ supports anaerobic respiration, while _TusA_ is essential for iron-sulfur cluster biogenesis [100]. _GlpE_ facilitates

**Table 6. Phenotypic antibiotic susceptibility testing results for *A. baumannii* DUEMBL6, determined using standard disk diffusion according to CLSI guidelines.**

| Antibiotic classes | Antibiotics | Susceptibility |
|---|---|---|
| Fluoroquinolones | Ciprofloxacin (5 µg) | Resistance |
| | Levofloxacin (5 µg) | Resistance |
| Aminoglycosides | Gentamicin (10 µg) | Resistance |
| | Tobramycin (10 µg) | Resistance |
| | Kanamycin (10 µg) | Resistance |
| Carbapenems | Imipenem (10 µg) | Resistance |
| | Meropenem (10 µg) | Resistance |
| Tetracyclines | Tetracycline (30 µg) | Resistance |
| | Doxycycline (30 µg) | Resistance |
| Phenicols | Chloramphenicol (30 µg) | Resistance |
| Cephalosporins | Ceftazidime (30 µg) | Resistance |
| | Cefepime (30 µg) | Resistance |
| | Cefotaxime (30 µg) | Resistance |
| Macrolides | Azithromycin (30 µg) | Resistance |
| Penicillins | Ampicillin (10 µg) | Resistance |

sulfur detoxification, and *CorC* and *CorA* mediate magnesium and cobalt efflux, maintaining cation homeostasis [97,100]. These genes enhance bacterial survival in metal- and hydrocarbon-contaminated environments, potentially contributing to cross-resistance between metals and antibiotics [101].

Since the bacterium has a large repertoire of antibiotic resistance genes and virulence genes that can promote persistence in the human host, we estimated the pathogenic potential using the PathogenFinder. It predicts a probability of 86.1% that the *A. baumannii* DUEMBL6 is a human pathogen, with 614 matches to known pathogenic families and none to non-pathogenic ones. While the *A. baumannii* DUEMBL6 might be a promising candidate for bioremediation, the pathogenic potential raises public health concerns during its application in environmental cleanup efforts. The use of bacteria with pathogenic potential in bioremediation raises concerns about the risk of spreading antibiotic resistance genes and causing infections. This is particularly relevant for *A. baumannii*, given its history of hospital outbreaks and ability to survive on surfaces [102]. To mitigate these risks, future work should evaluate strategies such as employing non-pathogenic, functionally redundant microbial consortia in place of pathogenic isolates, genetically attenuating virulence factors or resistance determinants to reduce pathogenicity, and using immobilization or encapsulation techniques to confine bacterial activity and prevent uncontrolled environmental spread. Incorporating such biosafety measures would be essential before considering field deployment of *A. baumannii* or related isolates for bioremediation.

## Conclusion

The study underscores the efficacy of indigenous bacterial isolates, particularly *A. baumannii* DUEMBL6, in diesel degradation, driven by enzymatic pathways like alkane hydroxylation and aromatic ring cleavage. While *A. baumannii* DUEMBL6's genomic repertoire (e.g., *alkB* mutations, siderophore production) enhances its adaptability to hydrocarbon-rich environments, its multidrug resistance and virulence traits pose significant challenges for safe bioremediation deployment. It is important to note that conclusions regarding the functional advantage of the observed *alkB* mutations (I138F, A334T) remain speculative. Future research should therefore include enzyme activity assays, transcriptomics, and proteomics to experimentally validate their role in hydrocarbon degradation, as well as targeted deletion or inactivation of key genes to confirm their contribution to biodegradation. In addition, engineering approaches to remove virulence and antibiotic

resistance determinants, and functional assays to test the activity of biosynthetic pathways, represent promising directions. Efforts should also prioritize the use of non-pathogenic, genetically stable strains or microbial consortia to mitigate ecological and health risks. This research contributes to the growing body of knowledge on microbial bioremediation in low-resource settings, emphasizing the need for balanced strategies that harness microbial potential while minimizing unintended consequences.

## Supporting information

**S1 Table. List of antimicrobial resistance genes detected in *A. baumannii* DUEMBL6.**
(PDF)

**S2 Table. List of virulence genes detected in *A. baumannii* DUEMBL6.**
(PDF)

## Author contributions

**Conceptualization:** Nizam Uddin, Monjima Islam Prova, Muttasim Billaha.

**Data curation:** Tasnimul Arabi Anik, Rahat Uzzaman.

**Funding acquisition:** Anowara Begum.

**Investigation:** Nizam Uddin, Monjima Islam Prova, Muttasim Billaha.

**Methodology:** Nizam Uddin, Monjima Islam Prova, Muttasim Billaha.

**Project administration:** Anowara Begum.

**Resources:** Humaira Akhter, Anowara Begum.

**Supervision:** Anowara Begum.

**Visualization:** Tasnimul Arabi Anik, Faruk Islam.

**Writing – original draft:** Tasnimul Arabi Anik.

**Writing – review & editing:** Tasnimul Arabi Anik, Rahat Uzzaman, Nadia Haider.

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
