## [Editor Report · Decision Letter 0]

30 May 2025

Dear Dr. Begum,

We look forward to receiving your revised manuscript.

Kind regards,

Ranjit Gurav, Ph.D

Academic Editor

PLOS ONE

“The research was funded by the Ministry of Science and Technology, and University Grant Commission, Bangladesh.”

“The research was funded by the Ministry of Science and Technology, and University Grant Commission, Bangladesh.”

“The research was funded by the Ministry of Science and Technology, and University Grant Commission, Bangladesh.”

Additional Editor Comments:

Thank you for your submission. Upon reviewing the manuscript, we found that Figures 1 to 7 are not sufficiently clear in their current form. To proceed with the review process, we kindly request that you resubmit the manuscript with high-resolution versions of these figures to ensure their clarity and legibility for the reviewers.

---

## [Author Response · Author response to Decision Letter 1]

10 Jun 2025

1.Manuscript Formatting:

We have revised our manuscript according to PLOS ONE’s style requirements, using the formatting templates provided for both the main body and the title/author/affiliations sections.

2.Role of the Funders:

The funding statement has been updated to clarify the role of funders. Please amend the Funding Disclosure to read:

“The research was funded by the Ministry of Science and Technology, and University Grant Commission, Bangladesh. The funders had no role in study design, data collection and analysis, decision to publish, or preparation of the manuscript.”

3.Funding Statement Location:

We have removed any funding-related information from the Acknowledgments and other manuscript sections. The Funding Statement remains exclusively in the appropriate section of the submission form.

4.Supporting Information Captions:

Captions for all Supporting Information files have now been included at the end of the manuscript, and corresponding in-text citations have been updated accordingly to meet the journal's guidelines.

5.High-Resolution Figures:

The previous lower resolution may have resulted from the conversion to a combined PDF during initial submission. According to the journal, ‘The compiled submission PDF includes low-resolution preview images of the figures after the reference list. The function of these previews is to allow you to download the entire submission as quickly as possible. Click the link at the top of each preview page to download a high-resolution version of each figure.’ Yet, we have replaced some of the figures with high resolution versions.

---

## [Decision Letter · Decision Letter 1]

1 Oct 2025

Dear Dr. Begum,

Thank you for submitting your manuscript to PLOS ONE. After careful consideration, we feel that it has merit but does not fully meet PLOS ONE’s publication criteria as it currently stands. Therefore, we invite you to submit a revised version of the manuscript that addresses the points raised during the review process.

plosone@plos.org

Ranjit Gurav, Ph.D

Academic Editor

**Journal Requirements:**

Reviewers' comments:

**Comments to the Author**

2. Is the manuscript technically sound, and do the data support the conclusions?

Reviewer #1: Yes

Reviewer #2: Partly

Reviewer #3: Yes

3. Has the statistical analysis been performed appropriately and rigorously?

Reviewer #1: Yes

Reviewer #2: I Don't Know

Reviewer #3: Yes

4. Have the authors made all data underlying the findings in their manuscript fully available?

Reviewer #1: Yes

Reviewer #2: Yes

Reviewer #3: Yes

5. Is the manuscript presented in an intelligible fashion and written in standard English?

Reviewer #1: Yes

Reviewer #2: Yes

Reviewer #3: Yes

**Reviewer #1:** (No Response)

**Reviewer #2:** The authors report the isolation of bacteria from oil-contaminated sites in Dhaka, Bangladesh using 1% diesel enrichment/selection media. The authors isolated a range of bacteria (including Gammaproteobacteria) from this environment, many of which would be predicted based on previous literature. The authors test biofilm formation and diesel utilization of the strains before choosing an isolate of Acinetobacter baumannii for genome sequencing and bioinformatic analysis. My specific comments are below:

1. The paper would benefit from further background discussion of the alkB and catE genes, their roles biodegradation, and previous uses in the field.

2. The authors mention comparison of % diesel degradation to other previously tested strains (lines 269-270), but these analyses were not done “side by side” as far as I can see. Were similar conditions used, so as to make this a valid comparison? Ideally, it would have been optimal to include other control/comparison strains in their analysis.

3. I did not see any figure legends in the material that was available for review. Please provide figure legends for the reader.

4. The authors discuss genes/relevant pathways that are present in the sequenced Acinetobacter isolate. The study would benefit from deletion/inactivation of some of these genes and then observing if this affects the biodegradation activity.

5. The authors discuss the presence of genes/systems for biosynthesis pathways in the sequenced isolate (as opposed to biodegradation). Is there any evidence that these pathways are functional? Could some assays be performed to test this?

6. The authors discuss the fact that virulence genes and antibiotic resistance genes were detected in the sequenced Acinetobacter isolate and that these genes could pose a problem for the use of the strain in biodegradation. The authors should suggest that these genes could possibly be deleted/inactivated in this strain, and then the resulting strain tested to see if this affects the biodegradation activity. In other words, the strain could possibly be engineered to eliminate virulence and antibiotic resistance for future use/applications. Experiments toward this end would improve the paper.

7. Was the sequenced isolate tested for relevant antibiotic resistance? This would improve the paper and give insight into whether those genes are expressed/functional.

8. Whenever the authors refer to the sequenced strain DUEMBL6, it would help the reader if they include the genus/species in key locations to help the reader (some examples: titles to Figures 4, 5, and 7; line 357, line 362, etc). I realize you mention this strain in the text, but it helps the reader overall when reading the whole paper to include the genus/species, especially since there are many strain numbers used in the paper (for example, write as “Acinetobacter baumannii DUEMBL6”).

**Reviewer #3:**  Dear Authors,

Thank you for the opportunity to review your manuscript titled “Genomic Analysis of Acinetobacter baumannii DUEMBL6 Reveals Diesel Bioremediation Potential and Biosafety Concerns.” The study is methodologically sound, timely, and relevant, but I believe it can be strengthened by addressing the following points:

1. Comparative Performance of DUEMBL6

o Please consider including a comparative framework, such as a table summarizing the diesel degradation efficiencies of DUEMBL6 alongside those of other globally reported strains or consortia. This will help contextualize your findings.

2. Biosafety and Application Risks

o The biosafety discussion requires more depth. I recommend elaborating on potential strategies such as the use of non-pathogenic microbial consortia, genetic attenuation of virulence factors, and immobilization or encapsulation techniques to prevent uncontrolled spread.

3. Functional Validation of Genomic Insights

o The alkB mutation analysis (I138F, A334T) is interesting, but conclusions regarding functional advantage remain speculative. Please acknowledge this as a limitation and recommend future work, such as enzyme activity assays, transcriptomics, and proteomics, to validate functionality.

4. Figures and Visuals

o A summary table of antibiotic resistance genes and virulence factors should be moved into the main text for ease of reference rather than being confined to supplementary materials.

5. Language and Formatting

o The manuscript would benefit from polishing for clarity and consistency. In particular, I noted several typographical errors that should be corrected:

a) Page 31, Line 306: “Idetification of biosynthetic gene clusters” → “Identification of biosynthetic gene clusters”

b) Page 36, Line 366: “Antibitotic resistance genes and mobile genetic elements” → “Antibiotic resistance genes and mobile genetic elements”

c) Page 20, Line 219 / Table 2: “Strenotrophomonas maltophila” → “Stenotrophomonas maltophilia”

With improvements in comparative analysis, biosafety framing, data presentation, and language/formatting, the manuscript will present its findings with much greater clarity and impact.

**Do you want your identity to be public for this peer review?** For information about this choice, including consent withdrawal, please see our Privacy Policy

Reviewer #1: No

Reviewer #2: No

Reviewer #3: No

---

## [Author Response · Author response to Decision Letter 2]

8 Oct 2025

REVIEWER 1

1.The paper would benefit from further background discussion of the alkB and catE genes, their roles biodegradation, and previous uses in the field.

Response to Comment 1:

We thank the reviewer for this insightful suggestion. In response, we have expanded the discussion of the alkB and catE genes to include both their established roles in hydrocarbon biodegradation and their previous applications in the field. Specifically, we now highlight that alkB-encoded alkane hydroxylases are widely used as biomarkers for alkane degradation in marine and soil environments, while catE-encoded catechol-2,3-dioxygenases play a central role in aromatic compound degradation and have been applied in the bioremediation of oil-contaminated soils and groundwater (van Beilen & Funhoff, 2007; Harayama et al., 1999; Wang et al., 2011; Jiao et al., 2016). These additions are included in the Results and Discussion section (Lines 251-255).

2.The authors mention comparison of % diesel degradation to other previously tested strains (lines 269-270), but these analyses were not done “side by side” as far as I can see. Were similar conditions used, so as to make this a valid comparison? Ideally, it would have been optimal to include other control/comparison strains in their analysis.

Response to Comment 2:

We thank the reviewer for this valuable comment. Since different studies were conducted under varying experimental conditions, we have included the respective conditions in Table 3 for proper context. It was not possible to identify studies performed under exactly identical parameters, as a study may have preferential temperatures and conditions based on their region or specific goals. Therefore, the comparisons presented here are intended to provide contextual rather than experimental insights into the relative degradation efficiency of our isolates. This clarification has been added to the revised manuscript (Lines 260–266, Table 3).

3.I did not see any figure legends in the material that was available for review. Please provide figure legends for the reader.

Response to comment 3

We thank the reviewer for this suggestion. Figure legends have now been added to accompany all figures to provide clarity for the reader.

4. The authors discuss genes/relevant pathways that are present in the sequenced Acinetobacter isolate. The study would benefit from deletion/inactivation of some of these genes and then observing if this affects the biodegradation activity.

5. The authors discuss the presence of genes/systems for biosynthesis pathways in the sequenced isolate (as opposed to biodegradation). Is there any evidence that these pathways are functional? Could some assays be performed to test this?

6. The authors discuss the fact that virulence genes and antibiotic resistance genes were detected in the sequenced Acinetobacter isolate and that these genes could pose a problem for the use of the strain in biodegradation. The authors should suggest that these genes could possibly be deleted/inactivated in this strain, and then the resulting strain tested to see if this affects the biodegradation activity. In other words, the strain could possibly be engineered to eliminate virulence and antibiotic resistance for future use/applications. Experiments toward this end would improve the paper.

Response to comments 4, 5, and 6:

We thank the reviewer for these insightful suggestions. The primary scope of the present study was to identify and characterize the isolate and to provide a genomic overview of its biodegradation potential. We agree that functional validation through gene deletion/inactivation studies (comment 4), biochemical assays to test the activity of biosynthetic pathways (comment 5), and engineering approaches to remove virulence and antibiotic resistance determinants (comment 6) would substantially strengthen the understanding and potential application of this strain. We have now acknowledged these points in the Conclusion (lines 503-508) as important directions for future work. At present, we are seeking funding support to pursue these experimental approaches, and we believe that the publication of this study will help us justify and build the foundation for such follow-up research.

7.Was the sequenced isolate tested for relevant antibiotic resistance? This would improve the paper and give insight into whether those genes are expressed/functional.

Response to comment 7

We thank the reviewer for this valuable suggestion. We had previously performed phenotypic antibiotic resistance testing, but initially excluded these results to keep the focus on genomic characteristics and avoid excessive length. As suggested, we have now incorporated these data and linked the genomic resistance determinants to the corresponding phenotypic profiles (Lines 198-204, 419-421, 424-426, 434-436, and 442-443).

8.Whenever the authors refer to the sequenced strain DUEMBL6, it would help the reader if they include the genus/species in key locations to help the reader (some examples: titles to Figures 4, 5, and 7; line 357, line 362, etc). I realize you mention this strain in the text, but it helps the reader overall when reading the whole paper to include the genus/species, especially since there are many strain numbers used in the paper (for example, write as “Acinetobacter baumannii DUEMBL6”).

Response to comment 8

We thank the reviewer for this valuable suggestion. We have revised the manuscript to include the genus and species name (e.g., Acinetobacter baumannii DUEMBL6) in the specified figure titles (Figures 4, 5, and 7) and key text locations (lines 357, 362, etc.). We believe this change improves clarity and consistency for readers, especially given the presence of multiple strain numbers in the manuscript.

REVIEWER 3

1.Comparative Performance of DUEMBL6

o Please consider including a comparative framework, such as a table summarizing the diesel degradation efficiencies of DUEMBL6 alongside those of other globally reported strains or consortia. This will help contextualize your findings.

Response to Comment 1:

We thank the reviewer for this thoughtful suggestion. In response, we have included a comparative framework (Table 3) summarizing the diesel degradation efficiencies of A. baumannii DUEMBL6 alongside other globally reported strains and consortia. Since these studies were conducted under different environmental and experimental conditions, we have also included their respective parameters (e.g., temperature, incubation period) to ensure proper contextualization. The comparison is intended to provide perspective on the relative degradation efficiency of DUEMBL6 rather than a direct experimental equivalence. These additions can be found in the revised manuscript (Lines 260–266, Table 3).

2.Biosafety and Application Risks

The biosafety discussion requires more depth. I recommend elaborating on potential strategies such as the use of non-pathogenic microbial consortia, genetic attenuation of virulence factors, and immobilization or encapsulation techniques to prevent uncontrolled spread.

Response to comment 2

We thank the reviewer for this important suggestion. We agree that the biosafety aspects of applying A. baumannii DUEMBL6 in bioremediation require more depth. We have now expanded the Discussion to highlight potential strategies to mitigate these risks, including the use of non-pathogenic microbial consortia, genetic attenuation of virulence and resistance factors, and immobilization or encapsulation techniques to prevent uncontrolled environmental spread (Lines 490–495). We believe this addition strengthens the manuscript by addressing biosafety considerations and providing a framework for future applied research.

3. Functional Validation of Genomic Insights

The alkB mutation analysis (I138F, A334T) is interesting, but conclusions regarding functional advantage remain speculative. Please acknowledge this as a limitation and recommend future work, such as enzyme activity assays, transcriptomics, and proteomics, to validate functionality.

Response to comment 3

We thank the reviewer for this valuable suggestion. We have acknowledged the speculative nature of the alkB mutation analysis as a limitation and have recommended future validation through enzyme activity assays, transcriptomics, and proteomics in the Conclusion (Lines 501–505).

4. Figures and Visuals

A summary table of antibiotic resistance genes and virulence factors should be moved into the main text for ease of reference rather than being confined to supplementary materials.

Response to comment 4

We appreciate the reviewer’s suggestion. However, the summary tables of antibiotic resistance genes and virulence factors span more than one page, and per the journal’s formatting requirements, such large tables must remain in the supplementary materials. Therefore, we have kept these tables in the supplementary file.

5. Language and Formatting

o The manuscript would benefit from polishing for clarity and consistency. In particular, I noted several typographical errors that should be corrected:

a) Page 31, Line 306: “Idetification of biosynthetic gene clusters” → “Identification of biosynthetic gene clusters”

b) Page 36, Line 366: “Antibitotic resistance genes and mobile genetic elements” → “Antibiotic resistance genes and mobile genetic elements”

c) Page 20, Line 219 / Table 2: “Strenotrophomonas maltophila” → “Stenotrophomonas maltophilia”

Response to comment 5

We thank the reviewer for pointing out these errors. The typographical and formatting issues noted (Pages 20, 31, and 36) have been corrected in the revised manuscript.

---

## [Decision Letter · Decision Letter 2]

18 Nov 2025

Dear Dr. Begum,

Thank you for submitting your manuscript to PLOS ONE. After careful consideration, we feel that it has merit but does not fully meet PLOS ONE’s publication criteria as it currently stands. Therefore, we invite you to submit a revised version of the manuscript that addresses the points raised during the review process.

As per Reviewer 2 comments, the manuscript discusses virulence and antibiotic resistance genes but lacks experimental validation of their functional roles. The authors were encouraged to consider gene deletion or inactivation studies targeting key virulence and antibiotic resistance determinants in future work. Such experiments would offer valuable insights into the genetic basis of resistance and virulence and could demonstrate the feasibility of engineering strains with reduced pathogenicity and resistance. However, it is understandable that these experiments may not be feasible within the current revision timeline.

Reviewer 2 also noted that the authors had mentioned performing antibiotic resistance assays in the previous revision. Upon review, it was observed that while the assay has been described in the Methods section and discussed in the Results, the corresponding data (e.g., figures, tables, or supplementary material) are missing. Therefore, it is recommended to revise the manuscript and incorporate the missing antibiotic resistance assay data (figures or tables) to support their discussion in the manuscript.

We look forward to receiving your revised manuscript.

Kind regards,

Ranjit Gurav, Ph.D

Academic Editor

PLOS ONE

Journal Requirements:

Reviewers' comments:

Reviewer's Responses to Questions

**Comments to the Author**

Reviewer #1: All comments have been addressed

Reviewer #2: (No Response)

Reviewer #3: All comments have been addressed

2. Is the manuscript technically sound, and do the data support the conclusions?

Reviewer #1: Yes

Reviewer #2: Yes

Reviewer #3: (No Response)

3. Has the statistical analysis been performed appropriately and rigorously?

Reviewer #1: Yes

Reviewer #2: I Don't Know

Reviewer #3: (No Response)

4. Have the authors made all data underlying the findings in their manuscript fully available?

Reviewer #1: Yes

Reviewer #2: Yes

Reviewer #3: (No Response)

5. Is the manuscript presented in an intelligible fashion and written in standard English?

Reviewer #1: Yes

Reviewer #2: Yes

Reviewer #3: (No Response)

Reviewer #1: All comments from the reviewers have been addressed accordingly by the authors. Congratulations on the work done.

Reviewer #2: (No Response)

Reviewer #3: (No Response)

**Do you want your identity to be public for this peer review?** For information about this choice, including consent withdrawal, please see our Privacy Policy

Reviewer #1: No

Reviewer #2: No

Reviewer #3: **Yes:** JONATHAN AKWASI ADUSEI

---

## [Author Response · Author response to Decision Letter 3]

24 Nov 2025

As per Reviewer 2 comments, the manuscript discusses virulence and antibiotic resistance genes but lacks experimental validation of their functional roles. The authors were encouraged to consider gene deletion or inactivation studies targeting key virulence and antibiotic resistance determinants in future work. Such experiments would offer valuable insights into the genetic basis of resistance and virulence and could demonstrate the feasibility of engineering strains with reduced pathogenicity and resistance. However, it is understandable that these experiments may not be feasible within the current revision timeline.

Response to Comment 1

We thank the reviewer for this constructive comment. We agree that functional validation through gene deletion or inactivation studies would provide deeper insights into the roles of key virulence and antibiotic resistance determinants. While these experiments are beyond the scope of the current revision, we previously added this point to the conclusion as a prospect for future work. We believe this addition not only acknowledges the limitation but also highlights a clear direction for extending the findings of this study.

Reviewer 2 also noted that the authors had mentioned performing antibiotic resistance assays in the previous revision. Upon review, it was observed that while the assay has been described in the Methods section and discussed in the Results, the corresponding data (e.g., figures, tables, or supplementary material) are missing. Therefore, it is recommended to revise the manuscript and incorporate the missing antibiotic resistance assay data (figures or tables) to support their discussion in the manuscript.

Response to Comment 2

We thank the reviewer for pointing this out. We have now incorporated the missing antibiotic resistance assay data in Table 5 (Line 428) to support the corresponding descriptions in the Methods and Results sections. This table summarizes the phenotypic susceptibility profile.

---

## [Editor Report · Decision Letter 3]

9 Dec 2025

Genomic Analysis of Acinetobacter baumannii DUEMBL6 Reveals Diesel Bioremediation Potential and Biosafety Concerns

PONE-D-25-23823R3

Dear Professor

We’re pleased to inform you that your manuscript has been judged scientifically suitable for publication and will be formally accepted for publication once it meets all outstanding technical requirements.

Kind regards,

Ranjit Gurav, Ph.D

Academic Editor

PLOS One

---

## [Editor Report · Acceptance letter]

PONE-D-25-23823R3

PLOS One

Dear Dr. Begum,

I'm pleased to inform you that your manuscript has been deemed suitable for publication in PLOS One. Congratulations! Your manuscript is now being handed over to our production team.

Kind regards,

on behalf of

Dr. Ranjit Gurav

Academic Editor

PLOS One